# Use of GnRH-Encapsulated Chitosan Nanoparticles as an Alternative to eCG for Induction of Estrus and Ovulation during Non-Breeding Season in Sheep

**DOI:** 10.3390/biology12030351

**Published:** 2023-02-22

**Authors:** Nesrein M. Hashem, Ahmed S. El-Hawy, Moharram F. El-Bassiony, Ibrahim S. Abd El-Hamid, Antonio Gonzalez-Bulnes, Paula Martinez-Ros

**Affiliations:** 1Department of Animal and Fish Production, Agriculture Faculty (El-Shatby), Alexandria University, Alexandria 21545, Egypt; 2Animal and Poultry Physiology Department, Desert Research Center (DRC), Cairo 11753, Egypt; 3Departamento de Produccion y Sanidad Animal, Facultad de Veterinaria, Universidad Cardenal, Herrera-CEU, CEU Universities, C/Tirant lo Blanc, 7, Alfara del Patriarca, 46115 Valencia, Spain

**Keywords:** nano delivery system, gonadotrophins, seasonal anestrous, sheep, reproduction

## Abstract

**Simple Summary:**

Sheep are seasonal polyestrus breeders, and applying intensive breeding systems requires effective estrus induction protocols. Commonly, progesterone-equine chorionic gonadotrophin (eCG) protocols are used to induce estrus and ovulation in anestrous sheep. However, eCG is an essential element in such protocols, and it has many disadvantages such as the formation of antibodies against the hormone and aspects related to animal welfare (specifically pregnant mares). Gonadotrophin releasing hormone (GnRH) can be a possible eCG alternative, but variable ovarian responses and sexual behaviors were observed, and these were mainly due to the ability of GnRH to induce a precocious LH surge. Hence, optimizing the pharmacokinetics and pharmacodynamics of GnRH may meet the ovarian and behavioral requirements of ewes during the follicular phase. In this study, we compare the effects of two doses of GnRH-encapsulated chitosan nanoparticles as a new delivery system, with eCG in a progesterone-based estrus induction protocol, on ovarian response, progesterone concentrations, and pregnancy outcomes. The results indicate that, despite the fact that the high dose of nanoencapsulated GnRH shortened the duration of estrus, it improved progesterone concentrations during early pregnancy and lambing and fecundity rates compared with low nanoencapsulated GnRH and eCG treatments. Thus, a high dose of nanoencapsulated GnRH can be used as an eCG alternative in progesterone-based protocols in seasonally anestrous ewes.

**Abstract:**

This study is aimed at determining the reproductive performance of anestrous ewes treated with nanoencapsulated GnRH after a progesterone-based protocol for estrus induction was proposed as a way of replacing eCG. A total of sixty anestrous, multiparous, non-lactating Barki ewes were randomly allocated into three homogenous groups and subjected to a CIDR-based estrus induction protocol. The first group (eCG) received an intramuscular (i.m.) injection of 350 IU of eCG at CIDR removal. The second (LNGnRH) and third (HLNGnR) groups received either 25 µg or 50 µg of encapsulated GnRH nanoparticles by the i.m. route in the form of spherical GnRH-encapsulated chitosan–TPP nanoparticles (which were 490.8 nm and had a 13.6 mV positive charge) 48 h after CIDR removal. Follicular dynamics, estrous behavior, luteal activity, and pregnancy outcomes were evaluated. Three days after CIDR removal, the number of large follicles increased by similar amounts in the LNGnRH and eCG groups and were significantly higher in both groups than in the HNGnRH group. However, no differences were observed in the numbers and diameters of CLs among the experimental groups and, on the other hand, treatment with HNGnRH significantly increased blood serum progesterone levels compared with eCG and LNGnRH. Treatment with HNGnRH increased conception, lambing, and fecundity rates (*p* < 0.05), with the trend of a higher litter size (*p* = 0.081) compared with eCG, whereas LNGnRH resulted in intermediate values. In conclusion, a dose of 50 µg of GnRH encapsulated in chitosan-TPP nanoparticles can be used as an alternative to eCG in progesterone-based estrus induction protocols in sheep.

## 1. Introduction

In sheep production, the maximization of farm profit and sustainability can be achieved by applying intensive breeding systems to avoid non-productive periods (i.e., to obtain three lambing seasons in two years). This reproductive system, however, requires the application of suitable estrus induction protocols for inducing breeding activity during anestrous periods, as sheep are seasonal polyestrus breeders [1]. Worldwide, protocols based on the administration of progesterone and equine chorionic gonadotrophin (eCG) are commonly used to induce estrus and ovulation in sheep during both breeding and non-breeding seasons. Such protocols depend on a progesterone source to compensate for the absence of active corpus lutea due to a lack of ovulation, and they depend on eCG to stimulate folliculogenesis and estrous behavior upon the removal of the progesterone sponge/device [2]. Thus, eCG is an essential element in estrus induction protocols during non-breeding seasons and is highly recommended for designing an effective estrus induction protocol for different seasons.

However, currently, the production of eCG from pregnant equine species has raised serious ethical concerns because it interferes with animal welfare standards, as the extraction and manufacture of the hormone depends on the extensive bleeding of pregnant mares [3]. According to these aspects, there is a rising trend to replace eCG with other gonadotrophins. Among the different gonadotrophins, gonadotrophin releasing hormone (GnRH) represents a possible option since GnRH analogues can induce the internal release of pituitary gonadotrophins, follicle stimulating hormone (FSH), and luteinizing hormone (LH), and therefore stimulate folliculogenesis, ovulation, and luteinization [4]. Several studies have pointed to the possibility of the use of GnRH as an alternative to eCG in estrous synchronization and estrus induction protocols in sheep. However, the results related to estrous behavior and fertility rates after the use of GnRH as an eCG alternative, particularly in progesterone-based induction protocols in sheep, are inconclusive [5,6]. As inferred from previous published studies on the use of GnRH as an eCG alternative, the main challenge of the use of GnRH is related to its strong tendency to induce a precocious LH surge, which in turns induces ovulation and impedes estrous behavior or terminates it earlier than the eCG-based protocols do, affecting preovulatory follicle growth [7]. 

Hence, there is a need to slow down the effects of exogenous GnRH, in this sense, the administration of GnRH in propylene glycol [7]. However, the first results were promising during breeding season but not during the breeding season, indicating the need to optimize the delivery system and therefore the release of endogenous LH.

Nanotechnology may change the pharmacokinetic properties of gonadotrophins used in different assisted reproductive techniques and consequently its biological effects [8]. The conjugation of a hormone with a suitable nanoparticle is hypothesized to increase its half-life, improve its passage across epithelial/endothelial barriers into blood or lymph circulation, and sustain hormone release [9]. Hence, the use of nanoencapsulated GnRH may be an adequate way to release the hormone. A previous study conducted in our laboratory has shown that GnRH-loaded chitosan–TPP nanoparticles have longer release profiles over 24 h, indicating the ability of such formulas to sustain the release of GnRH for a longer time and with gradual release [8] even in reduced doses, compared to GnRH free forms [10,11]. 

Hence, the current study hypothesizes that nanoencapsulated GnRH may be used as an alternative to eCG in common protocols for the induction of estrus and ovulation, by allowing adequate follicular wave development and maturation before ovulation. Therefore, in this study, we compare the reproductive response of anestrous sheep to traditional protocols that involve eCG and alternative protocols that involve different doses of nano-GnRH.

## 2. Materials and Methods

This study was carried out in collaboration between the laboratory of nanobiotechnology and microbiology at the Faculty of Agriculture of Alexandria University, Egypt, the Desert Research Center at Egypt, and the Research Group on Animal Health and Welfare (SABIA) of the Faculty of Veterinary Sciences, Universidad Cardenal Herrera—CEU, Spain. The research involving the reproductive management of animals was conducted in agreement with experimental procedures assessed and approved by the University Committee of Ethics in Animal Research (report CEEA17/019), according to the Spanish Policy for Animal Protection (RD53/2013), which meets the European Union Directive 2010/63/UE.

### 2.1. Preparation and Characterization of GnRH-Encapsulated Chitosan Nanoparticles

The ionic gelation method described by Hashem et al. [11] was used with some modifications. Briefly, GnRH (Ovurelin, 100 µg gonadorelin (as acetate), /mL, Bayer New Zealand Limited, Manukau, Auckland, New Zealand) was encapsulated by nanoparticles of chitosan (Cat No. AL1234 00100; Alpha Chemica, Maharashtra, India) and hardened with TPP (TPP; Thermo Fisher GmbH, Kandel, Germany) to obtain encapsulated GnRH nanoforms. The chitosan (0.1%, *w/v*) was vigorously stirred in an aqueous acidic solution (1%, *v/v*) to obtain chitosan cation nanoparticles. Furthermore, an aqueous solution of TPP (0.1%, *w/v*) was prepared. The chitosan solution was first mixed with GnRH (2:1) and then the mixture was slowly dropped into the TPP solution (3 chitosan–GnRH: 1 TPP) under constant magnetic stirring (1200 rpm) at 25 °C for 60 min. The formula was then subjected to physiochemical characterization to assess different physicochemical properties including the size of particles (nm), polydispersity (PdI), and surface charge (Zeta potential, mV) (Malvern ZETASIZER Nano series, Worcestershire, UK). The topographic shape of GnRH (Gonadorelin)-encapsulated chitosan–TPP nanoparticles was imaged by using a scanning electron microscope (SEM; Jeol JSM- 6360 LA, 3–1–2 Musashino, Akishima, Tokyo, Japan) after coating them with gold to improve the imaging of the encapsulated hormone sample. The encapsulation efficiency, EE, of chitosan–TPP nanoparticles for GnRH was determined. The samples (n = 3) of GnRH-encapsulated chitosan–TPP nanoparticles were subjected to centrifugation at 8000 rpm for 20 min. The absorbance of the supernatant containing free GnRH (non-encapsulated GnRH) were measured at 280 nm using a UV-spectrophotometer (OPTIZEN POP, LAB Keen Innovation Solution, Daejeon, Republic of Korea). The EE was calculated by using the equation EE = (initial GnRH concentration − concentration of free GnRH) × 100/initial GnRH concentration [10].

### 2.2. Experimental Design

The study involved 60 multiparous, non-lactating Barki ewes (2–5 years old with a mean body weight of 40.10 ± 1.07 kg) kept outdoors with access to indoor facilities at the experimental farm of the Desert Research Center (Matrouh, Egypt). The animals were disease-free and clinically normal and received their daily nutritional requirements according to NRC [12], with free access to water.

The study started in March (spring season), which is a late non-breeding season for Egypt [13]. The ewes were treated with a CIDR-based estrus induction protocol (CIDR^®^ Ovis, Zoetis, Madrid, Spain) for 12 days and afterwards randomly allocated into three homogenous groups (n = 20/group). The first group (eCG-group) received an intramuscular (i.m.) injection of 350 IU of eCG (Gonaser, Laboratorios Hipra, S.A., Girona, Spain) at CIDR removal. The second (LNGnRH) and third (HLNGnR) groups received either 25 µg or 50 µg of encapsulated GnRH nanoparticles in an i.m. injection at 48 h after CIDR removal.

### 2.3. Estrus Detection and Mating

Estrous behavior was checked four consecutive days after CIDR removal by using teaser rams fitted with marking crayons. The crayons were checked and changed for a new color every 12 h.

The ewes with crayon marks indicating the occurrence of estrus were recorded and immediately mated with fertile rams. Estrus rates, the onset of estrus, and the duration of estrus were calculated. The time of the onset of estrous behavior was estimated by recording the halfway point between the time of CIDR removal and the first time the ewes had their backs marked. The duration of estrus was estimated to be the halfway point between the first and the last time the ewes had their backs marked. 

### 2.4. Ovarian Activity

Follicle growth and luteal activity in response to the treatments were assessed by transrectal ultrasonography (Sonoscape A6 equipped with a 5.0/7.0 MHz linear array probe, AZoNetwork UK Ltd. NEO, Manchester, UK). At days 0, 2, and 3 of CIDR removal (i.e.: follicular phase of the induced estrus), the ovaries of 10 sheep in each group were scanned. In each scanning session, ≥2 mm follicles were recorded and classified into three follicular populations: small, ≥2–3 mm; medium, >3–<5 mm; and large (ovulatory) follicles, ≥5 mm [14].

Ovulation and luteal activity in response to the treatment was assessed at day 7 post-mating. Total number and size of each of the corpora lutea (CLs) were recorded. For progesterone analysis, a sample of 5 mL of blood was collected from each ewe and contained in a non-heparinized tube by means of jugular venipuncture. Progesterone concentrations were also assessed on days 7, 21, and 42 post-mating. In all the samples, sera were obtained by the centrifugation of blood samples at 2000× *g* for 20 min and stored at −20 °C. Blood serum progesterone concentration was measured using solid-phase enzyme immunoassay kits obtained from Monobind Inc., CA, USA. The lower limit of detection was 0.105 ng/mL of serum and the intra- and inter assay CVs were 9.3%, 3.1%, and 2.9% and 9.9%, 7.0%, and 5.6% for the low, normal, and high samples, respectively.

### 2.5. Reproductive Performance and Pregnancy Outcomes

Pregnancy was identified in all females subjected to mating by detecting the presence of a conceptus 35 days post-mating by transrectal ultrasonography.

The reproductive performance of ewes was evaluated by calculating the following variables: conception rate = (no. of ewes identified as pregnant on day 35/no. of ewes mated) × 100; lambing rate = (no. of ewes delivered/no. of ewes mated) × 100; litter size = (no. of lambs born/no. of ewes delivered); fecundity = (no. of lambs born/no. of ewes mated) × 100. The litter weight of each lamb at birth was also recorded.

### 2.6. Statistical Analysis

All data were analyzed using the SAS/STAT packages (version 9 edition. Cary, NC, USA: SAS Inst, Inc; 2004). The results were firstly subjected to a normality test. A normality analysis showed that all data were normally distributed except those on ovarian follicle distribution. Hence, a Kruskal–Wallis test and a one-way ANOVA nonparametric test were used to analyze the data on follicle distribution (number of total and small, medium, and large follicles). The data was collected once and followed normal distribution, and the numbers and diameters of corpora lutea, onset of estrus, duration of estrus, litter size, fecundity, and litter weight were subjected to a one-way ANOVA test using a generalized linear model (GLM). Least-squares procedures using a mixed model were used, with consideration of the time of data collection as repeated measurements, to assess the effect of estrus induction protocols on blood serum progesterone concentrations on days 7, 21 and 42 post-mating. Categorical data on estrus, conception, and lambing rates were analyzed by using a chi-square test (χ2 test). All results were shown as means (±SEM), except for categorical data, which were shown as percentages. Differences between the means of different experimental groups were detected using Duncan’s new multiple range test with a *p*-value of less than 0.05.

## 3. Results

### 3.1. Physicochemical Properties of GnRH-Encapsulated Chitosan–TPP Nanoparticles

Physicochemical properties and scanning electron microscope images of GnRH-encapsulated chitosan–TPP nanoparticles are shown in Figure 1A–D. The average size of GnRH-encapsulated chitosan–TPP nanoparticles was 490.8 nm, distributed in two peaks, and the majority, 94.5%, of nanoparticles were 430.0 nm. The average PdI value of GnRH-encapsulated chitosan–TPP nanoparticles was 0.528 (Figure 1A). The fabricated nanoparticles had a positive zeta potential value (13.6 mV, Figure 1B). The SEM of the GnRH-encapsulated chitosan–TPP nanoparticles showed spherical particle morphology (Figure 1C,D). The EE of chitosan–TPP for GnRH was 89.24% (±2.44).

### 3.2. Effect of Treatment on Ovarian Activity and Progesterone Concentrations

There were no significant differences in the total number of follicles or in follicle distribution among all the experimental groups during CIDR removal (day 0). Afterwards, the treaeCG increased the number of medium follicles on day 2 after CIDR removal and, on day 3, it increased the number of large follicles, with a decrease in the number of small and medium follicles (*p* < 0.05 for all). The treatment with LNGnRH significantly decreased the number of small follicles but increased the number of large follicles from day 2 to 3 after CIDR removal. The treatment with HNGnRH resulted in an opposite trend to that observed after the treatment with LNGnRH (Figure 2). Hence, the treatment with eCG increased (*p* < 0.05) the number of small and medium follicles and the total number of follicles compared with the treatments with LNGnRH and HNGnRH on day 2 after CIDR removal. On day 3 after CIDR removal and 24 h after GnRH administration, the number of large follicles was significantly increased in the eCG and LNGnRH groups but not in the HNGnRH group (Figure 2).

No differences were observed in the numbers and diameters of CLs among the experimental groups (Table 1). It is worthy of note that the treatment with HNGnRH significantly increased the overall concentration of blood serum progesterone during the luteal phase and during early pregnancy (days 7, 21, and 42 post-mating) compared with eCG and LNGnRH (Table 1 and Figure 3).

### 3.3. Effect of Treatment on Reproductive Performance and Pregnancy Outcomes

There were no differences in the percentages of animals showing sings of estrus and in the interval between CIDR removal and the onset of estrus among the experimental groups (Table 2). On the other hand, there were differences in the durations of estrus, the shortest duration being found in the treatment with HNGnRH (*p* < 0.05). The treatment with HNGnRH increased (*p* < 0.05) conception, lambing, and fecundity rates compared to that with eCG, whereas that with LNGnRH resulted in intermediate values. The treatment with HGnRH showed a trend of improving litter size compared with other treatments (*p* = 0.081; Table 2).

## 4. Discussion

Protocols based on the administration of exogenous progesterone (usually as a CIDR device) combined with that of eCG are commonly used for estrus induction in sheep during non-breeding seasons. However, despite the fact that these protocols are widely applied during non-breeding seasons, the use of eCG has many disadvantages. eCG evokes follicular growth but has a long half-life (around 40 h, persisting in the blood circulation for up to 10 days in cows [15]), which results in more scattered occurrences of estrus and less tighter ovulation times [6]. The long half-life of eCG and its repeated use have been related to decreased ovarian responses with time, mainly because of the formation of antibodies against the hormone [16]. However, the main problem with the use of eCG arises from the increasing debate on the procedure by which the hormone is obtained, since it is originally extracted from the serum of pregnant equines, which is in conflict with animal welfare standards [8]. These facts make finding eCG alternatives an imperative issue.

Our group developed a protocol using gonadotrophin-releasing hormone (GnRH) instead of eCG [6]. Such a protocol is useful during breeding seasons but unfeasible during non-breeding seasons, with less than half of the ewes showing estrus behavior. Our hypothesis is that a single GnRH injection could not stimulate adequate development and maturation of the preovulatory follicle in non-breeding seasons, because sustained LH release is necessary for final follicle maturation and for the onset of estrous behavior [17]. Hence, the efficiency of the treatment would be improved by slowing and extending the release of GnRH.

Previous studies have shown the possibility of changing the pharmacokinetic properties of gonadotrophins used in different assisted reproductive techniques, and consequently, of changing its biological effects by controlling the delivery system. Santos-Jimenez et al. [7] found that administration of GnRH in propylene glycol may constitute a successful alternative to traditional estrus synchronization protocols based on the administration of eCG in sheep, but again, this can only be applied during breeding seasons because yields obtained again during non-breeding seasons were disappointing (unpublished results). In another study, Kumura [18] developed a FSH formula that allows it to be injected as a single dose, instead of repeated FSH doses, for superovulation by administering FSH in aluminum hydroxide gel. This formula resulted in the growth of multiple follicles, ovulation, and the production of multiple embryos similar to those obtained following eCG injection. In this scenario, nanotechnologies using nano drug/hormone delivery systems may constitute a good alternative. Generally, nano-formulated hormones have a more efficient passage across epithelial or endothelial barriers, a longer half-life time and sustained release, and greater cellular uptake [8].

In this study, our group developed a nano-GnRH formula with definite physicochemical properties (size, 490.8 nm; PdI, 0.528; zeta potential, 13.6 mV; EE, 89.24%), which had positive effects on ovulation and luteal function in different animal species such as goats, sheep, and rabbits compared with the available commercial forms of GnRH [11,19]. Based on these facts, we tested the potential of GnRH encapsulated in nanocargo (chitosan–TPP) as a new delivery system to replace eCG in a progesterone (CIDR)-based estrus induction protocol. We used two doses of encapsulated GnRH nanoparticles. The first dose (50 µg/ewe) was the common recommended dose for boosting follicular growth and ovulation in small ruminants, while the second dose represented the half (25 µg/ewe) of the commonly recommended dose [10]. These doses were selected to understand the behavior of the hormone after being in nanoform. In this study, we aimed to sustain the release of GnRH and to avoid the earlier induction of LH surge to mimic the effect of eCG when incorporated in such protocols.

Control sheep for the current study, treated with eCG during CIDR removal, showed the typical response to eCG, with the growth of small and medium follicles up to day 2 and an increased number of large ovulatory follicles up to day 3 after CIDR removal, and with almost all of them showing estrus behavior. On the other hand, the administration of a high dose of nanoencapsulated GnRH 2 days after CIDR removal resulted in a sudden decrease in the number of large follicles 24 h later, which suggests that the treatment induced a faster LH surge and thus ovulation within short time of administration. In contrast, the administration of the low dose of nanoencapsulated GnRH did not decrease the number of large follicles, which may indicate that the strong LH-stimulating effect of GnRH is attenuated when GnRH is used in a lower dose. These hypotheses are supported by the fact that the duration of was longer in the groups treated with eCG and with low doses of nanoencapsulated GnRH than in the group treated with the high dose. The high dose of nanoencapsulated GnRH would terminate estrous behavior earlier by advancing the LH surge and ovulation time.

In the present study, in spite of the fact that estrus rates were similar among the groups (90–100%), the treatment with eCG resulted in the lowest lambing rate (38.8%). This finding is consistent with those of other studies on subtropical sheep during non-breeding seasons. Hashem et al. [13] found that the lambing rate of anestrous Rahmani ewes (a fat-tailed breed) was 41.67% after the induction of estrus with CIDR–eCG (500 IU). Similarly, De et al. [20] obtained a 57.57% lambing rate during non-breeding seasons in subtropical Malpura and Kheri breeds. Moreover, several studies indicated the possible negative effects of eCG on embryo survival specifically during the pre-implantation period, either when used for in vivo [21] or in vitro [22,23] assisted reproductive techniques. For example, eCG of a dosage used for superovulation and, to a lesser degree, of a dosage used for estrus synchronization can affect glycosidase activity in the genital tract of the Chios breed of ewes before the maternal recognition of pregnancy. This may negatively affect embryo survival and the maintenance of pregnancy [21].

The fact that, in the current study, the treatment with the high dose of nanoencapsulated GnRH resulted in the highest pregnancy outcomes (conception and fecundity rates) indicates the advantage of the use of nanoencapsulated GnRH as an eCG alternative. Moreover, as indicated by our results, HNGnRH significantly improved progesterone concentrations during the early luteal phase and during early pregnancy. Several studies have reported the positive effects of GnRH administration during the time of mating or during the early luteal phase on both the luteinization of LH-responsive follicles and the subsequent luteal function and consequently, on progesterone concentrations [10,14].

## 5. Conclusions

Nanoencapsulated GnRH using chitosan–TPP nanoparticles can be used in progesterone (CIDR)-based estrus induction protocols for sheep as an alternative to eCG. In this study, we observed different ovarian responses, estrous behavior, and fertility rates when two doses of nanoencapsulated GnRH were used, indicating changes in the pharmacokinetic properties and biological effects of nanoencapsulated GnRH and paving the way for a promising alternative to eCG. Further studies must be conducted to explore the effect of the dose, time of administration after CIDR removal, and the response of different breeds to hormone treatments that depend on nano delivery systems.

## Figures and Tables

**Figure 1 biology-12-00351-f001:**
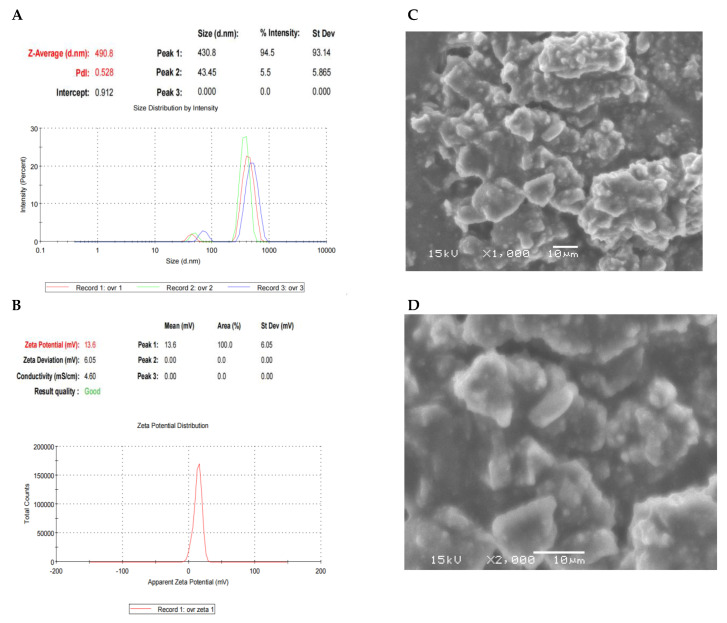
Physicochemical properties (size and polydispersity (**A**), zeta potential (**B**), and scanning electron microscope images of GnRH (Gonadorelin)-encapsulated chitosan–TPP nanoparticles at 1000× (**C**) and 2000× (**D**)).

**Figure 2 biology-12-00351-f002:**
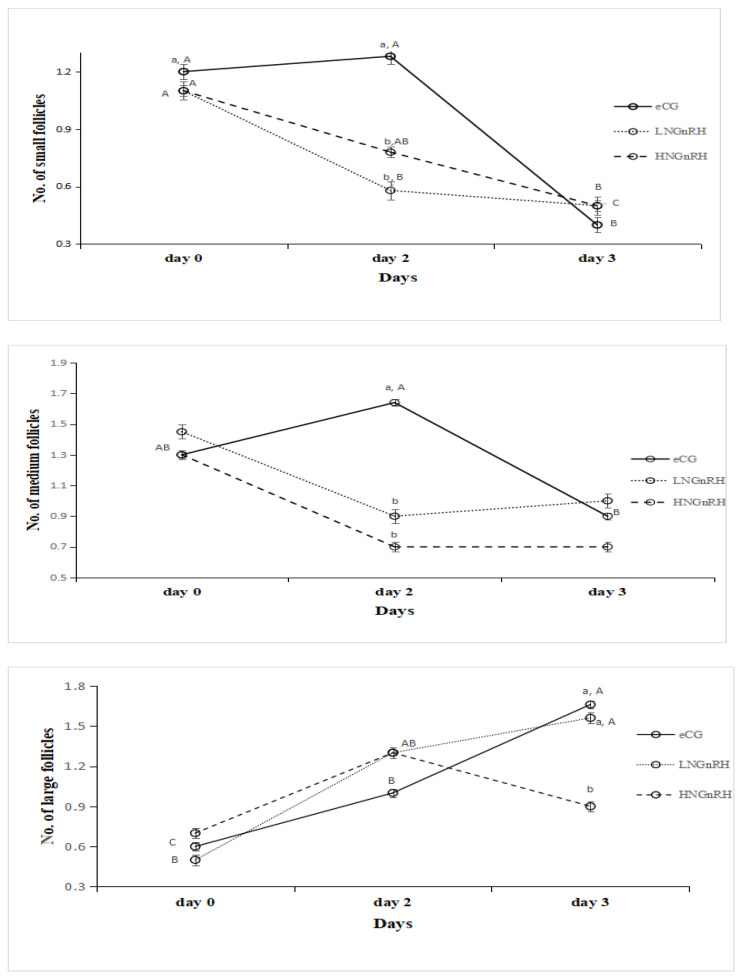
Follicular dynamics (small follicles: ≥2–3 mm, medium follicles: >3–<5 mm, and large follicles: ≥5 mm) of ewes treated with equine chorionic gonadotropin (eCG, 350 IU/ewe) during CIDR removal or under different doses of nano-encapsulated gonadotropin releasing hormone (LNGnRH, 25 µg/ewe and HNGnRH, 50 µg/ewe) 48 h after CIDR removal. Day 0: day of CIDR removal. The ^a,b^ means with different superscripts on the same scanning day are significantly different at *p* < 0.05. The ^A,B,C^ means with different superscripts in the same experimental group are significantly different at *p* < 0.05.

**Figure 3 biology-12-00351-f003:**
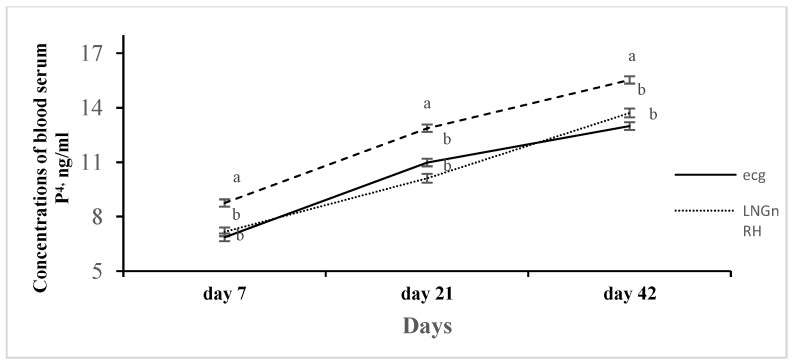
Concentrations of blood serum progesterone (P^4^, ng/mL), measured on day 7 (mid-luteal phase) and on days 21 and 42 (early pregnancy) post-mating of ewes treated with equine chorionic gonadotropin (eCG, 350 IU/ewe), during CIDR removal or under different doses of nano-encapsulated gonadotropin releasing hormone (LNGnRH, 25 µg/ewe and HNGnRH, 50 µg/ewe) 48 h after CIDR removal. The ^a,b^ means with different superscripts on the same analysis day are significantly different at *p* < 0.05.

**Table 1 biology-12-00351-t001:** Ovarian structures observed on day 7 post-mating and blood serum progesterone concentrations of ewes treated with equine chorionic gonadotropin (eCG, 350 IU/ewe) during CIDR removal or under different doses of nanoencapsulated gonadotropin releasing hormone (LNGnRH, 25 µg/ewe and HNGnRH, 50 µg/ewe) 48 h after CIDR removal (means ± standard error of means, SEM).

Item	eCG	LNGnRH	HNGnRH	SEM	*p*-Value
No. of corpora lutea	1.78	1.71	1.72	0.084	0.919
Diameter of corpus luteum	8.40	7.90	8.66	0.172	0.178
Progesterone ^1^, ng/mL	10.278 ^b^	10.325 ^b^	12.38 ^a^	0.611	<0.001

^1^ The overall mean concentrations of blood serum progesterone measured on day 7 (mid-luteal phase) and on days 21 and 42 (early pregnancy) post-mating. Changes among the days of determination are shown in Figure 3. The ^a,b^ means with different superscripts in the same row are significantly different at *p* < 0.05.

**Table 2 biology-12-00351-t002:** Estrus characteristics and pregnancy outcomes of ewes treated with equine chorionic gonadotropin (eCG, 350 IU/ewe) during CIDR removal under different doses of nano-encapsulated gonadotropin releasing hormone (LNGnRH, 25 µg/ewe and HNGnRH, 50 µg/ewe) 48 h after CIDR removal (Means ± standard error of means, SEM).

Item ^1^	eCG	LNGnRH	HNGnRH	SEM	*p*-Value
Estrous characteristics					
Estrous rate, %	90 (18/20)	100 (20/20)	100 (20/20)	-	0.912
Onset of estrus, h	33.6	35.2	37.75	1.91	0.404
Duration of estrus, h	28.22 ^a^	29.80 ^a^	13.60 ^b^	1.96	0.001
Pregnancy outcomes					
Conception rate on day 35, %	38.8 ^b^ (7/18)	65.0 ^ab^ (13/20)	80.0 ^a^ (16/20)	-	0.013
Lambing rate, %	38.8 ^b^ (7/18)	65.0 ^ab^ (13/20)	80.0 ^a^ (16/20)	-	0.013
Litter size	1.00 (7/7)	1.00 (13/13)	1.19 (19/16)	0.049	0.081
Fecundity,%	38.8 ^b^ (7/18)	65.0 ^ab^ (13/20)	95.0 ^a^ (19/20)	0.075	0.003
Birth weight, kg	2.66	2.61	2.51	0.043	0.364

^1^ Estrous rate = (no. of ewes who displayed estrus/no. of synchronized ewes) × 100%; onset of estrus = the halfway point between the time of CIDR removal and the first time the ewes had their backs marked; duration of estrus = the halfway point between the first time the ewes had their backs markedand the last one; conception rate = (no. of ewes identified as pregnant on day 35/no. of ewes mated) × 100; lambing rate = (no. of ewes delivered/no. of ewes mated) × 100%; litter size = no. of total lambs born/no. of ewes lambed; fecundity = (no. of lambs born /no. of ewes mated) × 100%. Numbers between brackets are numbers of animals. The ^a,b^ means with different superscripts in the same row are significantly different at *p* < 0.05.

## Data Availability

The data presented in this study are available on request from the corresponding author. The data are not publicly available because of privacy.

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
