# Peer review of "Use of GnRH-Encapsulated Chitosan Nanoparticles as an Alternative to eCG for Induction of Estrus and Ovulation during Non-Breeding Season in Sheep"

_biology, 2023, doi:10.3390/biology12030351_

Round 1

Reviewer 1 Report

1, the first group (eCG-group) received an intramuscular (i.m.) injection of 350 IU of eCG (Gonaser, Laboratorios Hipra, S.A., Girona, Spain) at CIDR removal. The second (LNGnRH) and third (HLNGnR) groups received either 25µg or 50µg of encapsulated GnRH nanoparticles in an i.m. injection at 48 h after CIDR removal.

 What is the theoretical basis of the experimental design (line 147-151)? what is purpose of each step of your design of experiment group?  It is better to be provided at the beginning of the discussion part.

2, if the sample ewes were more than 50 in each group with estrous rate, No. of corpora lutea, Diameter of corpus luteum and fecundity, the conclusion would be more accurate.

Author Response

1, the first group (eCG-group) received an intramuscular (i.m.) injection of 350 IU of eCG (Gonaser, Laboratorios Hipra, S.A., Girona, Spain) at CIDR removal. The second (LNGnRH) and third (HLNGnR) groups received either 25µg or 50µg of encapsulated GnRH nanoparticles in an i.m. injection at 48 h after CIDR removal.

 What is the theoretical basis of the experimental design (line 147-151)? what is purpose of each step of your design of experiment group?  It is better to be provided at the beginning of the discussion part.

We thank the comment from our reviewer and have thus added an additional part, please check lines 388-396:

We have compared the effects of two doses of encapsulated GnRH nanoparticles against a control group treated with eCG, which is the common protocol in sheep. First dose of GnRH nanoparticles (50µg/ewe) is the common recommended dose for boosting follicular growth and ovulation in small ruminants, while the second dose represents the half dose (25µg/ewe) of commonly recommended one (000). These doses were selected to understand the behaviour of the hormone after being in nanoform. In this study, we aimed to sustain the release of GnRH and to avoid earlier induction of LH-surge to mimick the effect of eCG when incorporated in such protocols.

2, if the sample ewes were more than 50 in each group with estrous rate, No. of corpora lutea, Diameter of corpus luteum and fecundity, the conclusion would be more accurate.

Thank you for your comment. Actually, we applied our treatments on 20 animals per group. This is the first study assessing GnRH effects in nanoform on reproductive performance of anestrous ewes, so we aimed to test our hypothesis first. Being obtained promising results, we can apply in larger numbers of animals as recommended.

Reviewer 2 Report

Comments: The manuscript entitled “Use of GnRH encapsulated chitosan nanoparticles as an alternative to eCG for induction of estrus and ovulation during non-breeding season in sheep by Nesrein M. Hashem et al. has determine the reproductive performance of anestrous ewes treated with nanoencapsulated GnRH following progesterone-based estrus induction protocol which replaced the eCG in response to the treatment. The major concerns are as follows:

1.     In Introduction (Lines 66-67), what animal welfare is affected by eCG? Please add details of its shortcomings.

2.     Line 137 (2.2. Experimental design), please check for formatting errors.

3.     Line 138-139, What effect will the different ages of the experimental animals have on the results of the experiment? Are these ewes primiparous or multiparous?

4.     Line 142, please check for punctuation errors.

5.     Line 160, Please check for errors.

6.     Line 167, Change " At days 0 2," to " At days 0, 2,".

7.     Line 171, Please keep the references consistent with the full text.

8.     In 2.4. Ovarian activity, why were only 10 ovaries of sheep in each group scanned, but not 20?

9.     Line 178, Change "2000×g" to "2000 ×g".

10.  Line 189-190, Please check the related errors.

11.  Lines 243, 246, 283…... and Table 1, P should be italicized.

12.  Line 267-268, Is there any duplication in the meaning expressed by the author? Please check for errors.

13.  Figure 3 is poorly visualized, please be consistent with Figure 1.

14.  Line 386-387, Please add the relevant references.

15.  Line 387, Change "De et al" to "De et al.".

16.  Please double check the writing and punctuation!

Author Response

Comments: The manuscript entitled “Use of GnRH encapsulated chitosan nanoparticles as an alternative to eCG for induction of estrus and ovulation during non-breeding season in sheep” by Nesrein M. Hashem et al. has determine the reproductive performance of anestrous ewes treated with nanoencapsulated GnRH following progesterone-based estrus induction protocol which replaced the eCG in response to the treatment. The major concerns are as follows:

  1. In Introduction (Lines 66-67), what animal welfare is affected by eCG? Please add details of its shortcomings.

Issues concerning animal welfare are not related to sheep receiving the treatment but to the mares in which the hormone is obtained. More details were added for explaining this fact. Please, check the following lines 69-71:

However, currently, the production of eCG from pregnant equine species has raised serious ethical concerns because it interferes with animal,s welfare standards, as the extraction and manufacture of the hormone depends on extensive bleeding of pregnant mares

  1. Line 137 (2.2. Experimental design), please check for formatting errors.

Thank you, adjusted as recommended.

  1. Line 138-139, What effect will the different ages of the experimental animals have on the results of the experiment? Are these ewes primiparous or multiparous?

The average of animals age is shown and all animals were multiparous. All animals were divided homogeneously on groups. Please, check lines 143-144.

  1. Line 142, please check for punctuation errors.

Corrected

  1. Line 160, Please check for errors.

Corrected

  1. Line 167, Change " At days 0 2," to " At days 0, 2,".

Corrected

  1. Line 171, Please keep the references consistent with the full text.

Corrected

  1. In 2.4. Ovarian activity, why were only 10 ovaries of sheep in each group scanned, but not 20?

This is because scanning of ovaries in small ruminant is relatively hard and time consuming. So, we scanned the half number of animals to be accurate and easier in agreement with other previous similar studies

  1. Line 178, Change "2000×g" to "2000 ×g".

Corrected

  1. Line 189-190, Please check the related errors.

Done

  1. Lines 243, 246, 283…... and Table 1, P should be italicized.

Corrected in these lines and table and along the article.

  1. Line 267-268, Is there any duplication in the meaning expressed by the author? Please check for errors.

No duplication, we used capital and small letters to compare between the means within a treatment or between the means of treatments.

  1. Figure 3 is poorly visualized, please be consistent with Figure 1.

More visualized figure was provided.

  1. Line 386-387, Please add the relevant references.

Relevant reference was inserted.

  1. Line 387, Change "De et al" to "De et al.".

Corrected

  1. Please double check the writing and punctuation!

Checked along the article.

Reviewer 3 Report

In this manuscript, the authors tested the efficiency of encapsulated GnRH in  protocoles (progesterone-based) compared to eCG in order to propose an alternative. Their results indicated that, although High dose of nanoencapsulated GnRH resulted in shorter estrous duration, it did not prevent any estrous behavior and resulted in better overall reproduction efficiency. They suggest that encapsulated GnRH result in slower release of the hormone and could be used as an alternative for eCG, although it would require additional tuning to optimize the release.

The manuscript is well written, easy to read and include a thoroughly described methodology. Although the number of animals per group is limited, it was sufficient to provide statistical power, comparable to past similar published researches and consistent with in vivo models. The conclusions are consistent with the presented results.

My comment is minor. I would suggest to do an additional thorough revision of the manuscript to correct the few minors text errors.

For example:

Line 22, 'alternatives' instead of 'altrenative'

Line 37, adding 'received' before 'an intramuscluar'

Line 69, remove the 'a' after hormone

Line 104, replace the 'and' a the beginning of the line by ' , '

Line 160 and 162, 'ewe's' instead of 'ewe,s'

Line 167, add a ',' between 0 and 2

Line 177, add an s to sample

Author Response

In this manuscript, the authors tested the efficiency of encapsulated GnRH in  protocoles (progesterone-based) compared to eCG in order to propose an alternative. Their results indicated that, although High dose of nanoencapsulated GnRH resulted in shorter estrous duration, it did not prevent any estrous behavior and resulted in better overall reproduction efficiency. They suggest that encapsulated GnRH result in slower release of the hormone and could be used as an alternative for eCG, although it would require additional tuning to optimize the release.

The manuscript is well written, easy to read and include a thoroughly described methodology. Although the number of animals per group is limited, it was sufficient to provide statistical power, comparable to past similar published researches and consistent with in vivo models. The conclusions are consistent with the presented results.

My comment is minor. I would suggest to do an additional thorough revision of the manuscript to correct the few minors text errors.

For example:

Line 22, 'alternatives' instead of 'altrenative'

Corrected

Line 37, adding 'received' before 'an intramuscluar'

Corrected

Line 69, remove the 'a' after hormone

Removed

Line 104, replace the 'and' a the beginning of the line by ' , '

Corrected

Line 160 and 162, 'ewe's' instead of 'ewe,s'

Corrected

Line 167, add a ',' between 0 and 2

Corrected

Line 177, add an s to sample

Corrected

Round 2

Reviewer 2 Report

The author has addressed my concerns